# The Bovine Herpesvirus 1 Latency-Reactivation Cycle, a Chronic Problem in the Cattle Industry

**DOI:** 10.3390/v15020552

**Published:** 2023-02-16

**Authors:** Jeffery B. Ostler, Clinton Jones

**Affiliations:** Department of Veterinary Pathobiology, College of Veterinary Medicine, Oklahoma State University, Stillwater, OK 74078, USA

**Keywords:** bovine herpesvirus 1, latency/reactivation, glucocorticoid receptor, stress, Wnt, β-catenin

## Abstract

Bovine alphaherpesvirus 1 (BoHV-1) is a persistent and recurring disease that affects cattle worldwide. It is a major contributor to bovine respiratory disease and reproductive failure in the US. A major complication of BoHV-1 arises from the lifelong latent infection established in the sensory ganglia of the peripheral nervous system following acute infection. Lifelong latency is marked by periodic reactivation from latency that leads to virus transmission and transient immunosuppression. Physiological and environmental stress, along with hormone fluctuations, can drive virus reactivation from latency, allowing the virus to spread rapidly. This review discusses the mechanisms of the latency/reactivation cycle, with particular emphasis on how different hormones directly regulate BoHV-1 gene expression and productive infection. Glucocorticoids, including the synthetic corticosteroid dexamethasone, are major effectors of the stress response. Stress directly regulates BoHV-1 gene expression through multiple pathways, including β-catenin dependent Wnt signaling, and the glucocorticoid receptor. Related type 1 nuclear hormone receptors, the androgen and progesterone receptors, also drive BoHV-1 gene expression and productive infection. These receptors form feed-forward transcription loops with the stress-induced Krüppel-like transcription factors KLF4 and KLF15. Understanding these molecular pathways is critical for developing novel therapeutics designed to block reactivation and reduce virus spread and disease.

## 1. BoHV-1 Is a World-Wide Pathogen That Causes Several Diseases

Bovine alphaherpesvirus 1 (BoHV-1) is a member of the Alphaherpesvirinae subfamily and genus Varicellovirus [1,2]. The genome is 135.3 kilobase pairs (kbp) with a high guanine and cytosine (GC) content and is arranged as a class D herpesvirus genome [3]. The viral genome contains a unique long (U_L_, 104 kbp) and unique short (U_S_, 10 kbp) region with the latter flanked on both sides by inverted internal and external repeats [4] (Figure 1A). Approximately 70 open reading frames (ORFs) have been identified, most of which are functional homologues to genes present in other alphaherpesviruses [4,5,6]. 

All BoHV-1 strains belong to one single viral species but are divided into three subtypes based on antigenic and genomic analysis: BoHV-1.1, -1.2a, and -1.2b [7]. Subtype 1 virus isolates are the causative agents for infectious bovine rhinotracheitis (IBR) and are typically found in the respiratory tract and aborted fetuses. Subtype 1 strains are common in Europe and North and South America. Subtype 2a infections are associated with a broad range of clinical symptoms in the respiratory and genital tract, for example, infectious pustular vulvovaginitis (IPV), infectious balanoposthitis (IPB), and abortions [8]. Subtype 2a was prevalent in Brazil and Europe prior to 1970 [8]. Subtype 2b has been isolated in Australia and Europe and is associated with respiratory disease, IPV, and IPB but not abortions [8,9].

## 2. BoHV-1 Is an Important Pathogen in Cattle 

Natural BoHV-1 infection occurs by contact with the virus in mucosal membranes of the upper respiratory or genital tracts of cattle [10]. Virus entry into the respiratory tract can occur via aerosol or by direct contact with virus present in nasal secretions [11]. Genital transmission generally requires direct contact at mating; however, virus has been isolated from semen samples used for artificial insemination [8,12]. Two major syndromes are caused by BoHV-1 infections: (i) IBR which occurs in the respiratory tract and (ii) IPV (cows) or IPB (bulls) which occurs in the genital tract. IBR is associated with clinical symptoms including pyrexia, apathy, increased respiratory rate with persistent harsh cough, and anorexia. Adult dairy cows also exhibit a severe drop in milk production. Primary infection occurs in the nasal and tracheal turbinate, while mucopurulent discharge from the nostrils and eyes is associated with pustular lesions in the nasal mucosa and conjunctivitis [4,13,14]. Multiple factors influence the clinical symptoms and severity of the disease caused by BoHV-1: strain-specific virulence, infected tissue type, secondary bacterial infection, host age and resistance factors. 

Notably, BoHV-1 is the most frequently diagnosed cause of viral abortion in North American cattle [15]. Although current BoHV-1 modified live vaccines are generally effective at preventing disease in cattle, infection or vaccination with a modified live vaccine can also induce abortion storms in pregnant cows [15,16,17,18,19]. BoHV-1, including commercially available modified live vaccines, can cause abortion storms where 50% of cows in herds are affected. Published studies concluded that unbred heifers vaccinated with a “killed” BoHV-1 vaccine have fewer abnormal estrus cycles and higher pregnancy rates when compared to heifers vaccinated with a modified live BoHV-1 vaccine [15,16,17,18,19]. Furthermore, current modified live vaccines reactivate from latency with nearly the same frequency as virulent strains present in nature, which complicates the use of these vaccines in cow and calf herds.

BoHV-1 infection is a major risk factor for bovine respiratory disease (BRD), a poly-microbial disease that can culminate in mortality due to bacterial pneumonia. BRD is the most economically important disease affecting beef and dairy cattle, accounting for approximately 75% of morbidities and >50% of mortalities in feedlot cattle [20,21,22,23,24]. *Mannheimia haemolytica* (MH) is a commensal bacterium [25] that belongs to the normal flora in the upper respiratory tract of healthy ruminants [26]. This commensal relationship is disrupted following stress or co-infections [27]; consequently, MH becomes the predominant organism that causes bronchopneumonia in many BRD cases [28,29,30,31]. BoHV-1 infection frequently causes upper respiratory tract disease [14,32], and co-infection of calves with MH consistently causes life-threatening pneumonia [33]. BoHV-1 enhances interactions between the MH leukotoxin and bovine peripheral blood mononuclear cells, including neutrophils [34,35]. Acute BoHV-1 infection impairs cell-mediated immunity [36,37,38,39], CD8+ T cell recognition of infected cells [40,41,42,43], and CD4+ T cell functions. BoHV-1 infection also erodes mucosal surfaces of the upper respiratory tract, which promotes MH colonization in the lower respiratory tract [28,29,30,31]. Notably, a BoHV-1 entry protein is a BRD susceptibility gene for Holstein calves [44], supporting the premise this virus is an important BRD cofactor. 

## 3. BoHV-1 Acute Infection

Acute infection induces apoptosis, inflammation, and high levels of virus shedding [13,14]. Lesions may be present, but subclinical viral shedding may also occur. Viral gene expression occurs in three well-defined phases: immediate early (IE), early (E), or late (L). IE gene expression begins immediately following virus entry and requires only host factors and virus tegument proteins present in the infecting virion. IE transcription unit 1 (IEtu1) encodes two transcriptional regulatory proteins (bICP0 and bICP4) which stimulate the productive infection [45,46,47] (Figure 1A). IEtu2 encodes the bICP22 protein [45]. These proteins are required for efficient viral gene expression. A tegument protein, VP16, stimulates IE transcription by interacting with sequences present in all IE promoters [48,49]. IE proteins drive E gene expression, which are generally translated into non-structural proteins that promote viral DNA replication. L proteins comprise the infectious virus capsid and tegument, which are required for virion assembly and egress. Viral particles produced during acute infection enter the peripheral nervous system via cell-cell spread into sensory neurons. A productive infection will periodically reoccur (reactivation from latency) as virus is shed from neurons periodically during the lifetime of the host.

**Figure 1 viruses-15-00552-f001:**
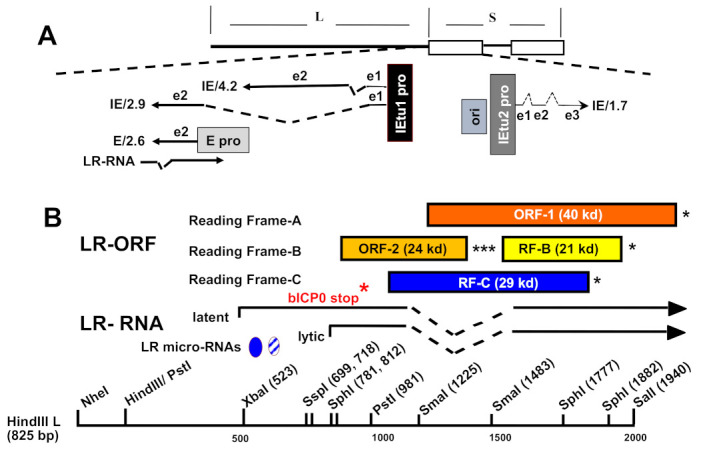
Schematic of BoHV-1 genes encompassing the LR gene. (**A**) The BoHV-1 genome contains a unique long (L), a unique short (S), and two repeats denoted by white rectangles. Positions of IE transcripts and LR transcript (LR-RNA) [45,46,50] are denoted. IE/4.2 is the IE transcript that encodes bICP4. IE/2.9 is the IE transcript that encodes bICP0. The IEtu1 promoter activates the expression of IE/4.2 and IE/2.9 and is denoted by a black rectangle (IEtu1 pro). E/2.6 is the early transcript that encodes bICP0, and a separate early promoter regulates the expression of the E/2.6 transcript (E pro). Exon 2 (e2) of bICP0 contains all the protein-coding sequences. The origin of replication (ORI) separates IEtu1 from IEtu2. The IEtu2 promoter (IEtu2 pro) drives the expression of the bICP22 protein. Solid lines in the transcript denote exons (e1, e2, e3) and dashed lines introns. (**B**) Partial restriction map of LR gene. The LR gene contains two open reading frames (ORF-1 and ORF-2) [51,52,53]. Reading frame B (RF-B) and reading frame C (RF-C) do not contain an initiating methionine. Asterisks denote the positions of in-frame stop codons. Start sties for lytic and latent LR-RNA are denoted. The position of the bICP0 termination codon (single red *) and LR-micro-RNAs are also denoted. The *** denote the position of the three in frame stop codons for ORF2.

## 4. BoHV-1 Establishes Life-Long Latency in Neurons and Lymphoid Tissue

If acute infection is initiated within the oral, nasal, or ocular cavity, sensory neurons in trigeminal ganglia (TG) are a primary site for latency [51,52,53,54]. In acutely infected sensory neurons, lytic cycle viral gene expression and virus production occur [55]. Lytic cycle viral gene expression is then extinguished, a significant number of infected neurons survive, and surviving infected neurons harbor viral genomes; these steps are operationally defined as the establishment of latency. The maintenance of latency is defined as long-term repression of lytic cycle viral gene expression and virus production. The only viral gene abundantly expressed during latency is the latency-related (LR) gene [52,53,56,57]; LR gene functions are discussed below. 

Pharyngeal tonsils (PT) are continually exposed to viral and bacterial pathogens. For example, viruses that infect the nasal cavity drain into PT and can potentially infect PT cells. Furthermore, the lacrimal duct system transmits passage of tear film from ocular surfaces to the nasal cavity; consequently, the BoHV-1 can drain into the PT. BoHV-1 DNA is consistently detected in PT of latently infected cattle [58,59,60,61]. Viral DNA from other 𝛼-herpesvirinae subfamily members is also detected in the PT of their respective natural hosts. These include pseudorabies virus [62,63], equine herpesvirus 4 [64], canine herpesvirus 1 [65], and *human alphaherpesvirus 1* (HSV-1) but not HSV-2 [66]. For these studies, their respective hosts were not shedding detectable levels of infectious virus nor were they exhibiting disease symptoms suggesting they were latently infected. We previously demonstrated that BoHV-1 lytic cycle viral gene expression is detected in PT using in situ hybridization when latently infected calves were given an IV injection of the synthetic corticosteroid dexamethasone (DEX) [58]. Consequently, we suggest that virus shedding during reactivation from latency via PT is crucial for virus transmission. 

## 5. LR Gene Products Modulate the Latency-Reactivation Cycle

### 5.1. Identification and Characterization of Viral Gene Products Expressed during Latency

The only BoHV-1 gene abundantly expressed during latency is LR-RNA [67,68,69]. In situ hybridization revealed that most LR-RNA is localized in the nucleus of latently infected neurons; however, LR-RNA is also detected in the cytoplasm [67,68,70,71]. A subset of LR-RNA is polyadenylated and alternatively spliced [72,73]. The start site for LR-RNA during productive infection is 24 bases downstream of the TATA box (Figure 1B) [74]. A different start site is utilized during latency that is 200–300 nucleotides upstream of the lytic site [73]. The LR gene promoter is contained within a 980 bp PstI fragment. This promoter is more active in neuronal cells when compared to non-neuronal cells and promoter activity is downregulated by DEX [75,76]. 

Two open reading frames (ORF) (ORF1 and ORF2) and two reading frames without an initiating methionine (RF-B and RF-C) are present in the LR gene [67,68,69,70,71] (Figure 1B). LR-RNA splicing in TG of calves generates a poly A+ mRNA where ORF2 is intact in TG at day one after the infection of the calves [72,77,78]. Alternative splicing of LR-RNA in TG generates ORF2 coding sequences at the amino terminus whereas the C-terminus contains portions of ORF1 or RF-B coding sequences at 7 or 15 dpi, respectively [72,77,78]. Full-length ORF1 would also be expressed from an unspliced LR transcript. In sharp contrast to TG, LR-RNA splicing in productively infected bovine cells is different indicating neuronal-specific splicing occurs in TG of infected calves. 

ORF2 is a multi-functional protein and a master regulator of the latency-reactivation cycle. For example, ORF2 impairs apoptosis in transiently transfected mouse neuroblastoma cells (Neuro-2A) suggesting this function promotes survival of infected cells, including neurons [79]. Yeast two-hybrid studies revealed ORF2 interacts with cellular regulatory proteins, for example, Notch family members, β-catenin, and CCAAT enhancer binding protein alpha (c/EBP-α) [80,81,82,83,84]. These interactions were confirmed in transiently transfected Neuro-2A cells. Interestingly, Neuro-2A cells consistently support ORF2 protein expression and expression of ORF2 isoforms generated by splicing from an expression vector; conversely, other cell types examined do not support protein expression. C/EBP-α and Notch-1 activate bICP0 expression, an important viral trans-activator [85,86], suggesting ORF2 impairs promoter activity via interactions with C/EBP-α and Notch1. The ability of Notch family members to impair neurite formation in Neuro-2A cells was also blocked when ORF2 was present [87], suggesting ORF2 enhances neuronal differentiation. Two LR-encoded micro-RNAs interfered with bICP0 protein expression [88] (see Figure 1B for locations of LR-encoded micro-RNAs). These viral micro-RNAs may also reduce expression of additional viral or cellular proteins. While ORF2 plays a prominent role in the BoHV-1 latency-reactivation cycle, other LR gene products are also predicted to be important. 

A mutant BoHV-1 virus that contains three stop codons at the beginning of ORF2 (LR mutant virus) does not express ORF2 or RF-C but expresses low levels of ORF1 [57,89,90]. Calf studies revealed that the LR mutant virus exhibits reduced virus shedding in ocular tissue, TG, and tonsils during acute infection. During establishment of latency, higher levels of apoptosis are detected in TG neurons of calves infected with the LR mutant when compared to calves infected with wt or the rescued LR mutant. This was unexpected because the LR mutant virus does not grow as well as wt BoHV-1 or the rescued LR mutant virus in calves. Strikingly, the LR mutant virus does not shed virus following DEX treatment, which triggers reactivation from latency in all calves infected with wt BoHV-1 or the rescued LR mutant [57,60]. Thus, ORF2 expression correlates with efficient establishment and maintenance of latency [91]. 

A recombinant human alphaherpesvirus 1 (HSV-1) that contains the BoHV-1 LR promoter and coding sequences inserted into an HSV-1 mutant lacking sequences encoding the first 1.5 kb of the latency-associated transcript (LAT) was constructed; this virus is referred to as CJLAT. CJLAT exhibits enhanced reactivation from latency in mice and rabbits when compared to the parental wild-type HSV-1 McKrae strain [92]. Notably, CJLAT also increases the incidence of fatal encephalitis in mice and rabbits relative to wild-type HSV-1. The introduction of three stop codons at the N-terminus of ORF2 and following insertion into the same site of the HSV-1 LAT mutant generated a virus that behaved like the parental LAT null mutant, suggesting ORF2 expression enhances reactivation from latency and pathogenesis [93]. Additional studies revealed CJLAT sustains corneal scarring and neovascularization after acute infection of rabbits, suggesting recurrent disease occurred [94]. These studies indicate that the multi-functional ORF2 enhances the HSV-1 latency-reactivation cycle by co-opting certain important cellular regulatory proteins and signaling pathways. HSV-1 LAT is predicted to express multiple micro-RNAs, two small non-coding RNAs, and a long-coding RNA but no functional promoter [51,52]. Consequently, ORF2 expression may have induced immune responses during acute infection and latency, which exacerbates ocular scarring. Finally, ORF2 expression may also enhance inflammation in the nervous system, culminating in encephalitis.

### 5.2. DEX Promotes BoHV-1 Productive Infection and Consistently Initiates Reactivation from Latency 

DEX, a synthetic corticosteroid hormone that mimics the effects of stress, consistently induces reactivation from latency in calves or rabbits latently infected with BoHV-1 [57,68,85,95,96,97,98,99,100,101]. Stressful stimuli activate the hypothalamic-pituitary-adrenal axis leading to the release of glucocorticoids (GCs) from the adrenal gland, reviewed in [102]. GCs have a pleiotropic effect on cell growth and survival, development, and metabolism. GCs enter a cell and bind to a cytoplasmic localized glucocorticoid receptor (GR) or a mineralocorticoid receptor (MR), reviewed in [103]. MR or GR homodimers bound to DEX are translocated to the nucleus where they bind consensus GR response elements (GRE), remodel chromatin, and induce transcription in the absence of de novo protein synthesis. Consequently, stress is an immediate early response that does not require de novo protein synthesis and stimulates transcription within minutes. DEX promotes productive infection in bovine kidney or Neuro-2A cells transfected with BoHV-1 genomic DNA [104,105]. This is independent of virion VP16 because viral DNA was transfected into cells; thus, the virion tegument containing VP16 was not initially present in the transfected cell. VP16 is expressed early during reactivation but is not initially present. 

Corticosteroids also modulate the serum and glucocorticoid-regulated protein kinases (SGK), members of the AGC family of serine/threonine protein kinases, reviewed in [106]. SGK1 mRNA and protein levels are rapidly increased following stressful stimuli. Inhibition of SGK activity demonstrated a marked reduction in BoHV-1 productive infection and viral protein synthesis [107]. Importantly, SGK inhibition did not block GR and DEX-dependent activation of the BoHV-1 IEtu1. This demonstrates multiple independent roles for stress-induced viral replication. 

During the early stages of DEX-induced reactivation in TG, bICP0 and VP16 are detected before bICP4 and bICP22 [108,109,110]. Conversely, bICP4 mRNA, but not bICP0, bICP22, and VP16, is abundantly expressed within 30 min after DEX treatment in PT of latently infected calves [111]. Since DEX reproducibly initiates reactivation from latency in the natural host, we have searched for viral and cellular factors that mediate reactivation from latency, as discussed below.

## 6. Regulation of Wnt/β-Catenin Signaling Pathway during Latency and DEX-Induced Reactivation

RNA sequencing studies revealed that more than 100 genes linked to the canonical Wnt/β-catenin signaling pathway are differentially expressed in TG during latency when compared to 30 min and 3 h after an intravenous DEX injection was given to latently infected calves to initiate reactivation from latency [81]. The canonical Wnt/β-catenin signaling pathway is more active during latency when compared to TG from uninfected calves or DEX-induced reactivation from latency (Figure 2A). Notably, four Wnt agonists are expressed at significantly higher levels during latency. The two genes with the highest levels of differential expression are the guanine nucleotide-binding protein alpha Q (GNAQ) [112,113] and Wnt ligand 16 (Wnt16). The third Wnt agonist is bone morphogenetic protein receptor 2 (BMPR2), a serine/threonine protein kinase, required for dorsoventral patterning of TG, peripheral innervation, and survival of sensory neurons [114,115,116]. Finally, a serine/threonine cellular protein kinase (Akt3) is differentially expressed [117] (see Figure 2A). Akt3 is more important than Akt1 and Akt2 with respect to blocking stroke-induced neuronal injury [118], blocking neuronal apoptosis, and triggering axonal development [117]. These differentially expressed cellular genes promote β-catenin stabilization and nuclear localization of β-catenin, in part because Wnt 16 or other Wnt proteins enhance expression of Disheveled, which impairs the association of β-catenin with the destruction pathway. The β-catenin destruction pathway is comprised of GSK-3b (glycogen 3 kinase 3b), Axin, APC (Adenomatous Polyposis Coli), and CKIa (Casein kinase I isoform alpha). Nuclear β-catenin forms an active transcription complex with T-cell factor/lymphoid enhancer factor (TCF/LEF) [119,120]. Generally, the canonical Wnt/β-catenin signaling pathway promotes cellular growth and constitutively activated β-catenin is linked to tumor cell development [121]. In sharp contrast, the canonical Wnt/β-catenin signaling pathway enhances neurogenesis and neuronal survival in the nervous system [120,121,122,123]. 

Recent studies revealed Akt1 and Akt 2 impair stress-induced transcription, whereas Akt3 enhances neurite formation in Neuro-2A [124]. Akt1 and Akt2 are expressed in TG during the latency-reactivation cycle, but their RNA levels do not change dramatically after DEX treatment. Furthermore, ORF2 stimulates β-catenin-dependent transcription in transfected cells [125] and promotes neuronal survival during latency. Finally, over-expression of β-catenin and a small non-coding RNA spanning part of ORF-2 coding sequences impair stress-induced transcription [126]. Collectively, these studies revealed that maintaining latency is an active process mediated, in part, by activation of the canonical Wnt/β-catenin signaling pathway. 

Expression of five known Wnt antagonists, dickkopf-1 (DKK1), dickkopf-1 like protein (DKKL1), Wnt inhibitory factor 1 (Wif-1), secreted frizzled-related protein 4 (SFRP4), SFRP5, and two intracellular Wnt inhibitors (SOX3 and SOX18) are induced within 3 h after DEX treatment to trigger reactivation from latency (Figure 2B). The canonical Wnt/β-catenin signaling pathway is negatively regulated by the DKK family of secreted proteins. In general, Wnt antagonists specifically bind to Wnt co-receptors, the low-density lipoprotein receptor-related proteins (LRP), and these interactions block Wnt family members from binding to the receptor/coreceptor complex [121,127,128]. SFRP family members bind to Wnt family members and block the Wnt pathway. Interestingly, SFRP4 also induces apoptosis [129,130], in part by blocking Akt signaling pathways [131]. A destruction complex interacts with β-catenin in the cytoplasm, culminating in reduced β-catenin steady-state protein levels (Figure 2B). Initially, β-catenin is phosphorylated by CKIa and then GSK-3β phosphorylates β-catenin. β-catenin phosphorylation targets the protein for ubiquitination and degradation by the proteasome. SOX3 and SOX18 impair β-catenin-dependent transcription independent of the β-catenin destruction pathway [132]. In summary, soluble Wnt antagonists induced by corticosteroids, including DEX, stabilize the β-catenin degradation complex and promote β-catenin phosphorylation and degradation. Induction of apoptosis, via impairment of the Wnt/Akt signaling pathways, may increase α-herpesvirus replication [133] and reactivation from latency [134]. 

## 7. GR and Stress-Induced Transcription Factors Stimulate Key Viral Promoters 

GR interactions with target DNA are mediated in part by the glucocorticoid response element (GRE, 5′-GGTACANNNTGTTCT-3′). Over 100 potential GREs were identified in the BoHV-1 genome, including two in the immediate early transcription unit 1 (IEtu1) promoter, a regulatory element that drives ICP4 and ICP0 expression following virus entry into a cell and at the onset reactivation (Figure 3A). These critical regulatory proteins drive viral gene expression and are required for efficient infection and reactivation from latency. Indeed, GR and DEX strongly activate the IEtu1 promoter, and transactivation is significantly reduced when either GRE is mutated [104]. Interestingly, one of the two GREs is an exact match to the consensus sequence, and mutating this site shows a more dramatic reduction in GR-induced activation relative to the other GRE, which contains a single mismatch from the consensus. Mutating both sites completely abolishes GR and DEX-induced promoter activation. 

During early stages of reactivation from latency, novel stress-induced cellular transcription factors were identified in TG [96], suggesting these cellular transcription factors play an important role during the reactivation from latency. These stress-induced transcription factors include promyelocytic leukemia zinc finger (PLZF), Slug, SPDEF (Sam-pointed domain containing Ets transcription factor), Krüppel-like transcription factor 4 (KLF4), KLF6, KLF15, and GATA6. These stress-induced transcription factors stimulate productive infection and key viral promoters [96,105,135,136,137,138,139]. The finding that 4 KLF family members (KLF4, KLF6, KLF15, and PLZF) are stimulated during DEX-induced reactivation from latency is important because KLF family members belong to the Sp1 transcription factor family and certain KLF family members interact with GC-rich motifs, including Sp1 binding sites, reviewed in [140,141]. The BoHV-1 genome is GC-rich, and many viral promoters contain Sp1 consensus binding sites and other GC-rich motifs, suggesting KLF binding sites are frequently found in the viral genome [140]. Expression of key viral regulatory proteins (bICP0, bICP4, bICP22, and VP16), but not glycoprotein C or E, are detected in TG neurons within 90 min after DEX treatment [108,109,110].

GR interacts with several of these stress-induced transcription factors forming feed-forward regulatory networks to enhance BoHV-1 gene expression. KLF4 and KLF15 are of particular importance. GR and KLF15 cooperatively transactivate the IEtu1 promoter in the presence of DEX by 40-fold, which is significantly higher than GR + DEX (~8-fold) or KLF15 alone (~3-fold) [139]. Transactivation is dependent upon the two GREs located in the DEX-responsive region of the IEtu1 promoter. As with GR + DEX, the consensus GRE is largely responsible for GR and KLF15-mediated transactivation, as a mutation of this sequence significantly reduces promoter activity. Additionally, a KLF-like binding site is present in this region; however, mutating this site only results in an approximate 30% reduction in promoter activity. It is notable that GR and KLF15 directly interact, as demonstrated by coimmunoprecipitation, suggesting cooperative transactivation is the result of GR-KLF15 protein complexes. 

The BoHV-1 genome was analyzed for similar motifs with GREs and a KLF-like site, and 13 sites were identified. Not all these sites were in promoters, but GR-dependent transactivation has been documented in genes up to 5kb from the GRE [142]. These 13 sites were analyzed for GR and KLF15-dependent transactivation. Interestingly, none of the sites were strongly transactivated by either GR + DEX or KLF15. However, four of these sites were significantly transactivated by GR and KLF15 cooperatively in the presence of DEX. These sites included a UL52 intergenic sequence, a bICP4 sequence, the IEtu2, and part of the unique short region. Of these, the UL52, bICP4 site, and IEtu1 are also transactivated by GR in combination with the related KLF4 protein. The UL52 intergenic sequence was strongly transactivated by GR and KLF15. UL52 is the primase subunit of the herpesvirus helicase-primase complex; how this GR-KLF15 responsive region affects virus gene expression or productive infection is unknown. 

IEtu1 drives bICP0 and bICP4 expression during the immediate early stages of BoHV-1 infection from a single primary transcript that is alternatively spliced (Figure 1). Additionally, a second promoter (bICP0 E pro) drives bICP0 expression during the early stage of infection [143]. This promoter is strongly transactivated by GR along with KLF4 and KLF15; however, DEX is either inhibitory, in the case of KLF4, or does not further stimulate activation in the case of KLF15 [105]. Ligand-independent GR activation is well documented, particularly with certain transcriptional coactivators [144,145]. The bICP0 E promoter is quite complex (Figure 3B). For example, it is highly GC-rich (~80%), contains numerous potential KLF and Sp1 binding sites, and has two half-GREs [105,138,139]. KLF4 and KLF15 belong to the Sp1/KLF superfamily of zinc-finger transcription factors. There is a similarity between Sp1 binding sites (GGGCGG) and certain KLF4 binding sites (GGGAGGG), and both KLF4 and KLF15 can bind Sp1 sites. KLF4 additionally binds to certain CACCC elements, reviewed in [140,141]. In the bICP0 E promoter, three separate and independent enhancer domains were identified, ranging from nucleotides 943 to 638 (EP-943), nucleotides 638 to 328 (EP-638), and nucleotides 328 to 172 (EP-328). Each domain is transactivated by GR and KLF4, while only EP-638 is transactivated by GR with KLF15 [105,146]. Both half-GREs are in the EP-943 fragment and are dispensable for transactivation. This is not surprising given the unliganded nature of GR-mediated transactivation of the bICP0 E promoter. However, an Sp1 and KLF4 site in this fragment are both required for transactivation by GR and KLF4. The EP-328 fragment also contains an Sp1 and a KLF4 site, both of which are required for transactivation. The EP-638 fragment contains two separate Sp1 sites, a KLF4 site, and a large triple Sp1 site (GGGCGGGGGCGGGGGCGGG). The triple Sp1 site is required for GR and KLF4 transactivation. These sites are directly responsible for GR and KLF4 mediated transactivation, as mutations reduced GR and KLF4 occupancy of enhancer fragments. It is notable that while GR expression plasmids were required for the transactivation of these enhancer fragments, chromatin immunoprecipitation (ChIP) experiments identified enhancer occupancy by endogenous GR, in the absence of over-expressing GR. Previous work demonstrated two potential isoforms of GR by western blot analysis, which migrate at approximately 90 and 120 kDa [139]. Only the 90kDa isoform is present in untransfected cells, while both are observed with GR overexpression. Since the expression vector only expresses GR-a, which migrates at 120 kd, it is clear GR-a, not the 90 kd GR protein, transactivates the bICP0 E promoter. Notably, both isoforms are detected following coimmunoprecipitation with KLF15 antibodies. The role that different GR isoforms play in the transactivation of BoHV-1 promoters is the subject of ongoing studies. 

## 8. Androgen and Progesterone Receptors Stimulate IE and E Promoters Cooperatively with KLF4 and KLF15

In addition to GR, the related androgen (AR) and progesterone receptors (PR) activate the BoHV-1 IEtu1 and bICP0 E promoters cooperatively with KLF4 and KLF15 [136,147,148]. AR and PR are type I nuclear hormone receptors activated by dihydrotestosterone (DHT) and progesterone (P4), respectively, allowing them to translocate to the nucleus and regulate transcription, similar to GR [149,150,151]. Importantly, both AR and PR recognize similar DNA sequences as GR and directly associate with KLF4 [147,148,152]. The AR strongly transactivates the IEtu1 promoter when stimulated with DHT, and this activation is increased by KLF15. In contrast, AR and KLF4 cooperatively transactivate the IEtu1 principally in the absence of DHT [147]. The PR also strongly activates the IEtu1 promoter when stimulated with P4, in cooperation with KLF15 [136]. IEtu1 transactivation by AR and PR is dependent upon two GREs, and their proximity to the transcription initiation site is important. When the 796 nucleotides between the TATA box and GREs are deleted (Figure 3A), cooperative transactivation between KLF15 and AR or PR is dramatically increased. The AR and PR also transactivate the bICP0 E promoter cooperatively with KLF4. As with GR, treatment with ligand (DHT or P4) impairs transactivation. The Sp1 and KLF4 binding sites in this promoter (Figure 3B) are required for KLF4 cooperative transactivation with AR and PR, suggesting a similar mechanism as GR. Furthermore, mutating these sites significantly reduced the occupancy of the promoter by all three transcription factors. 

The responsiveness of IEtu1 and bICP0 E promoters to stress and sex hormone signaling suggests these cellular signaling pathways mediate key aspects of the BoHV-1 life cycle in vivo. For example, stress or mating correlates with heightened virus transmission, suggesting an evolutionary response by BoHV-1. While it is not clear whether AR agonists (DHT for example) trigger reactivation from latency, P4 sporadically induces reactivation from latency in female calves [153] or rabbits [154] latently infected with BoHV-1. Collectively, these studies revealed sex hormones have the potential to stimulate viral gene expression and induce reactivation from latency. 

## Figures and Tables

**Figure 2 viruses-15-00552-f002:**
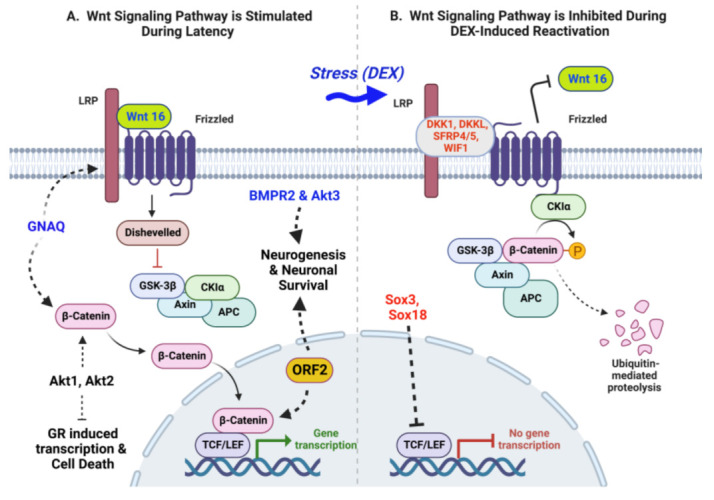
Summary of latency and DEX-induced reactivation on regulation of the canonical Wnt/β-catenin signaling pathway. (**A**) Key differentially expressed cellular genes identified in TG neurons during latency are shown in blue. (**B**) Summary of how DEX inhibits the Wnt/β-catenin signaling pathway in TG neurons, which correlates with reactivation from latency. Key cellular genes differentially expressed are shown in red. This schematic was generated using BioRender.

**Figure 3 viruses-15-00552-f003:**
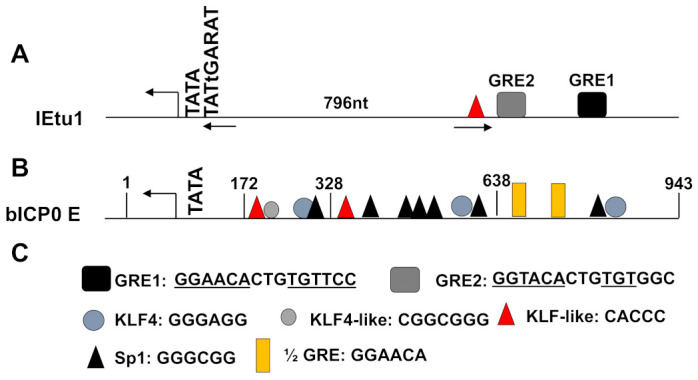
Schematic of BoHV-1 IEtu1 and bICP0 E promoters. (**A**) Key features of the IEtu1 promoter are depicted. The site of transcription initiation is indicated by an arrow immediately downstream of the TATA box and TATGARAT motif. The KLF-like CACCC and two GREs are also indicated. The TATA box is separated from GRE2 by 796 nucleotides. (**B**) Key features of the bICP0 E promoter are depicted. The site of transcription initiation is indicated by an arrow immediately downstream of the TATA box. The three enhancer domains are indicated: EP-328 (nucleotides 172 to 328), EP-638 (nucleotides 328 to 638), and EP-943 (nucleotides 638 to 943). Relative positions of Sp1, KLF4, and KLF-like sites are indicated. (**C**) Legend of transcription factor binding sites and consensus nucleotide sequences for binding DNA.

## Data Availability

The data is included in publications that were cited.

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
