# Peer review of "The Bovine Herpesvirus 1 Latency-Reactivation Cycle, a Chronic Problem in the Cattle Industry"

_viruses, 2023, doi:10.3390/v15020552_

Round 1

Reviewer 1 Report

This is a well written review of the latency-reactivation in cattle.  The review is well written and well structured. This review will be of great interest to general audience and to those who specifically work on alpha-herpes latency-reactivation. Although this review is interesting, there are several points that need to be addressed before publishing this review as follow:

  1. Please discuss cjLAT virus and how replacing bHSV LAT with HSV LAT affect the cycle of latency-reactivation any also why this virus more pathogenic. 
  2. Did the authors test the cJLAT virus in cattle?
  3. The authors needs to discuss the relevant of bHSV latency-reactivation to HSV latency-reactivation.
  4. Also, not important but it will be interesting if the authors discuss efficacy of bHSV vaccine against control of disease in cattle.

Author Response

With respect to the comments made by Reviewer 1, the following changes were made.
Concern #1: Please discuss cjLAT virus and how replacing bHSV LAT with HSV LAT affect the
cycle of latency-reactivation any also why this virus more pathogenic.
Response: This topic was briefly mentioned in the initial manuscript. This section was
expanded (lines 194-211).
Concern #2: Did the authors test the cJLAT virus in cattle?
Response: Considering CJLAT has gain of function in rabbits and mice, we did not feel
it was prudent to infect calves with this recombinant virus. I suspect this virus would have
escaped and could have been spread to humans.
Concern #3: The authors needs to discuss the relevant of bHSV latency-reactivation to HSV
latency-reactivation.
Response: Interesting point. However, there are many recent reviews focused on HSV-
1 latency and reactivation from latency. This review was written for a special edition in Viruses,
which is titled “Pathogenesis and Host Responses to Viral Diseases in Livestock Species”.
Hence, I did not feel it was appropriate to compare BoHV-1 to HSV-1.
Concern #4: Also, not important but it will be interesting if the authors discuss efficacy of bHSV
vaccine against control of disease in cattle.
Response: This section was expanded (lines 64-73).

Author Response

With respect to the comments made by Reviewer 3, the following changes were made.
Concern #1: The correct virus taxonomic nomenclature according to the ICTV should be used:
bovine alphaherpesvirus 1. The same should be considered for other viral species mentioned in
the text.
Response: Thank you for pointing this out. This was corrected for BoHV-1 (lines 11 and
31) and HSV-1 (line 194).
Concern #2: Figure 1. Panel A. The BoHV-1 genome contains a unique long (L), a unique
short region (S) region and two repeats… Delete ¨region¨ after (S) –
Response: this mistake was corrected (lines 109).
Concern #3: VP16….along the text is written as VP-16 or VP16. Use uniform spelling –
Response: VP-16 was changed to VP16. This was corrected throughout the manuscript
(line 316)
Concern #4: Line 225. expressed within 30 minutes after DEX treatment in pharyngeal tonsil
of calves latently infected calves. Delete ¨calves¨ before latently
Response: This was corrected (line 329).
Concern #5: Lines 243-244. These include the guanine nucleotide-binding protein alpha Q
(GNAQ) (1) and Wnt ligand 16 (Wnt16) are differentially expressed. Please revise this
sentence…it seems a word is missing.
Response: this sentence was modified, and I believe it is now easier to follow (lines
257-259).
Concern #6: Line 383. Previous work has demonstrated two potential isoforms of GR by
western blot analysis, which migrate at approximately 90 and 120 kDa. A reference for the
previous work should be provided.
Response: This reference is now added to that sentence (lines 396-398).

Reviewer 3 Report

The manuscript is interesting and well written. The topic has an enormous impact on cattle industry and trade movement. In this way the Authors contribute to highlight some still poorly untedstood aspects of the unique in vivo behaviour of alphaherpesviruses.

Author Response

Thank you.